# Effects of Sulfate and Sulfuric Acid on Efficiency of Geopolymers as Concrete Repair Materials

**DOI:** 10.3390/gels8010053

**Published:** 2022-01-12

**Authors:** Rayed Alyousef, Ahmed Abdel Khalek Ebid, Ghasan Fahim Huseien, Hossein Mohammadhosseini, Hisham Alabduljabbar, Shek Poi Ngian, Abdeliazim Mustafa Mohamed

**Affiliations:** 1Department of Civil Engineering, College of Engineering, Prince Sattam bin Abdulaziz University, Al-Kharj 11942, Saudi Arabia; bdggfh@nus.edu.sg (G.F.H.); h.alabduljabbar@psau.edu.sa (H.A.); a.bilal@psau.edu.sa (A.M.M.); 2Structural Engineering and Construction Management, Faculty of Engineering, Future University in Egypt, New Cairo 11835, Egypt; Ahmed.AbdelKhaleq@fue.edu.eg; 3Department of the Built Environment, College of Design and Engineering, National University of Singapore, Singapore 117566, Singapore; 4Institute for Smart Infrastructure and Innovative Construction (ISIIC), School of Civil Engineering, Universiti Teknologi Malaysia (UTM), Skudai 81310, Malaysia; mhossein@utm.my (H.M.); shekpoingian@utm.my (S.P.N.); 5Building and Construction Technology Department, Bayan University, Khartoum 11115, Sudan

**Keywords:** geopolymers, bond strength, compatibility, repair, durability, WCT, GBFS, FA

## Abstract

Various geopolymer mortars (GPMs) as concrete repairing materials have become effective owing to their eco-friendly properties. Geopolymer binders designed from agricultural and industrial wastes display interesting and useful mechanical performance. Based on this fact, this research (experimental) focuses on the feasibility of achieving a new GPM with improved mechanical properties and enhanced durability performance against the aggressive sulfuric acid and sulfate attacks. This new ternary blend of GPMs can be achieved by combining waste ceramic tiles (WCT), fly ash (FA) and ground blast furnace slag (GBFS) with appropriate proportions. These GPMs were designed from a high volume of WCT, FA, and GBFS to repair the damaged concretes existing in the construction sectors. Flexural strength, slant shear bond strength, and compatibility of the obtained GPMs were compared with the base or normal concrete (NC) before and after exposure to the aggressive environments. Tests including flexural four-point loading and thermal expansion coefficient were performed. These GPMs were prepared using a low concentration of alkaline activator solution with increasing levels of GBFS and FA replaced by WCT. The results showed that substitution of GBFS and FA by WCT in the GPMs could enhance their bond strength, mechanical characteristics, and durability performance when exposed to aggressive environments. In addition, with the increase in WCT contents from 50 to 70%, the bond strength performance of the GPMs was considerably enhanced under sulfuric acid and sulfate attack. The achieved GPMs were shown to be highly compatible with the concrete substrate and excellent binders for various civil engineering construction applications. It is affirmed that the proposed GPMs can efficiently be used as high-performance materials to repair damaged concrete surfaces.

## 1. Introduction

Geopolymer mortars (GPMs) as cement-free materials are useful for their high strength, excellent performance, and environmental friendliness. Most of the starting source materials abundant in calcium (CaO), aluminum (Al_2_O_3_), and silicon (SiO_2_) with alkali activation [1,2,3,4] are generally used to make these pastes, mortars, and concretes. Initial materials include palm oil fuel ash (POFA), fly ash (FA), granulated blast furnace slag (GBFS), etc. [5,6,7]. Many studies show that GPMs have a series of notable benefits, including fast setting time and curing at ambient temperatures [8,9]. Also, they have high early strength [10,11], high resistance to elevated temperatures, and strong durability in aggressive environments [12]; produce low levels of CO_2;_ and consume low amounts of energy [13,14]. Several researchers reported that the geopolymer performance was significantly influenced by binder chemical composition, alkaline solution modulus, solution to binder, and binder to aggregates ratios. Vasconcelos et al. [15] scrutinized the effect of different binder to aggregate ratios (0.3, 0.60, and 0.9) on compressive strength development of GPMs. The increase in strength appeared directly proportional to the increase in binder to aggregate ratio from 0.3 to 0.9. In [16], it was found that the optimum ratio of binder to aggregate (B:A) was 1:1; increasing binder content more than this ratio reduced the strength of GPMs. Hamzah et al. [17] reported the highest strength of GPMs was achieved with specimens’ mortar prepared with a ratio of 0.40 of alkaline solution to binder content at 28 days of age. Therefore, GPMs are well-suited and sustainable repair materials that can be applied as pastes for crack injections, mortar for section restoration, or patch repairs [18,19,20,21,22]. Several reports [23,24,25,26] suggested the preferred uses of FA alkali-activated mortars that include GBFS in the standard concrete repairing material. Compared to conventional cement-based repair materials, mortars made from FA and GBFS display higher shear bond and bending strength, indicating that they may be a favorable candidate to replace standard concretes in the repair work [26].

The ceramic industries worldwide produce huge quantities of wastes annually, and most end up in landfills. The reuse of such wastes in making the concretes can be beneficial in terms of both industrial waste problems’ solutions and environmental sustainability [27,28]. Annually, over 10 million square meters of produced ceramic tiles are wasted at global scale [29]. Some estimation showed that approximately 15–30% of the manufactured ceramics are wastes that are constantly accumulating [30] without being reused currently. This increasing percentage of ceramic wastes is mainly due to their damage while storing, transporting, and constructing or renovating buildings. In addition, the dumping procedure is also labor and money intensive because of growing stringent environmental regulations on landfilling. Therefore, there must be some alternate solutions wherein proper recycling or reuses of these waste ceramics into suitable construction products becomes feasible.

Repeated studies revealed that [28,30,31,32] ceramics are not biodegradable because they contain high amounts of crystalline silica and aluminum and are effective supplemental cementitious components for enhancing the mechanical and durability performance of concretes [33,34,35]. In spite of the great prospect of recycled ceramic wastes in the construction sector, a small amount of such ceramic wastes has been used as building materials so far [27,32]. Senthamarai et al. [28] evaluated the influence of the replacement of cement with ground WCT and showed the potential for the enhancement of the CS and durability performance, especially over longer curing durations owing to the ceramic waste’s pozzolanic characteristics. In a study by Samadi et al. [36], it was found that the inclusion of ceramic particles as both a cement replacement and fine aggregates in the mortar was highly effective in improving the durability performance, with a considerable increase in the sulfate resistance capacity of the mortar samples. Construction sectors being the main user of ceramic wastes may play a significant role in overcoming various environmental concerns. In fact, these wastes can safely be used to make GPMs without the need of any notable changes in the manufacturing processes and applications. Besides, the ceramic wastes dumping costs in landfills can be minimized in addition to less use of natural raw material resources, leading to increased energy saving and environmental sustainability. Several investigations affirmed that the construction industries can obtain more benefits in terms of sustainable development if they recycle various industrial wastes effectually as partial or full replacement of OPC to make geopolymer binders [10,37,38,39].

The term compatibility has become a popular buzzword in the repair industry. Generally, it implies durability of repair materials and adequate load-carrying capacity in the case of concrete repairs. It is, however, more than this. Compatibility can be defined as a balance of chemical, physical, and thermal properties and dimensions between a repair material (RM) and the NC that will ensure that the repair can withstand all the stresses induced by volume changes and chemical and thermal effects without distress and deterioration over a lifespan of designed specimens. In this view, several tests were adopted to evaluate the compatibility between the repair material and NC, which include coefficient thermal expansion (CTE), the flexural strength (FS) test for four-point loading, and the drying shrinkage (DS) test. The CTE of an RM is the measure of its length change when subjected to varying temperatures. When the RM and NC with different CTE are attached together and subjected to significant temperature changes, stresses are generated in the composite beam (GPMs-NC). Compatibility of concretes refers to how durable repairs are and whether they have the sufficient load-carrying capacity in terms of repair of concretes [40]. This essentially refers to the various chemical, electrochemical, and physical characteristics and dimensions of the repair materials. A compatible material is one in which the damaged substrates ensure that the repair puts up with the stress caused by the chemical reactions and volume changes in addition to electrochemical effects without showing distress or decay over a set time frame. This aspect is critical in terms of the concrete repairs, whereby the durability and sustainability of the repairs are influenced by the degree of bond compatibility [41,42]. The thermal expansion coefficient comprises a length change measurement shown in a material when it is exposed to varying temperatures. Stress is caused in composite materials if two materials possessing different thermal expansion coefficients are conjoined and then exposed to critical temperature change. Intensive studies have been conducted to determine the mechanical properties of diverse GPMs, especially their role in repairing the damaged concretes. To the best knowledge of the authors, very few studies have been conducted to assess the suitability of GPMs as concrete repair materials and evaluate the compatibility between GPMs and damaged cement-based concrete substrates. The majority of the earlier studies have focused on the assessment of mineralogy and the microstructural properties of GPMs.

In brief, an all-inclusive review of the existing literature revealed that the binary blend of GBFS and FA activated with high concentration alkaline solution is beneficial in improving the performance of GPMs. Most of the researchers are focused on evaluating the mechanical properties of GPMs. They depend on the compressive and bond strength as critical factors to evaluate GPMs as repair materials for repairing the concrete deterioration. There is a lack of information on the durability of GPMs as a repair material, such as compatibility between proposed mortar and concrete substrate and bond strength in aggressive environments. From the background, it is shown that the ternary blend content high volume WCT, FA, and GBFS activated with low sodium hydroxide molarity and sodium silicate content GPMs need to be developed. Considering the immense significance of industrial wastes-based GPMs, this study explored whether a new form of high-performance GPMs can efficiently be used to repair the damaged concrete surfaces because of its enhanced mechanical properties and high resistance against aggressive environmental attack. To meet this goal, a series of new GPMs were made using high volume of industrial wastes, including ceramics, GBFS, and FA at proper ratios. The measured thermal compatibility and bond strength of the developed GPMs (enclosing varied quantities of WCT as substitute to GBFS and FA) were compared with the base or normal concrete (NC) upon exposing them in sulfate and sulfuric acid environments. The thermal expansion coefficient, four-point load flexural, and slant shear bond strength tests were conducted to determine the bonding between GPMs and mortar or concrete substrates before and after the aggressive attacks. The new findings of this research are believed to render a basis for further studies and better knowledge on the behavior of a ternary-blend GPM obtainable from the waste material in a cheap and environmentally friendly manner.

## 2. Results and Discussion

### 2.1. Compressive Strength (CS)

Figure 1 shows the CS of the GPMs as a function of various WCT contents at 28 days of curing. Compared to the control sample (GPM_1_), the inclusion of 50% of WCT in place of FA in the GPMs was found to reduce their CS from 80.5 to 74.1 MPa. In addition, the CS of GPMs varied inversely with increase in WCT contents from 50 to 70%, wherein the values of CS were reduced from 70.1 to 34.8 MPa, respectively. The observed reduction in the mortars’ CS was mainly ascribed to the low calcium and high silica contents [10,43,44], which led to the formation of little C-(A)-S-H gels, thus lowering the CS. It is worth noting that WCT contained over 70% silica with larger grain size (35 µm) compared to GBFS and FA, which significantly affected the CS of GPMs designed with a high volume of WCT. GPMs made from a high volume of WCT attained a CS of 81% at 28 days of age. With increasing FA level in GPMs the amounts of silica and aluminum increased, which negatively influenced the calcium level (Ttableable 1). With the decrease in GBFS level from 50% (GPMs_2_) to 20% (GPMs_10_) the CS of GPMs was reduced from 70.1 to 22.4 MPa, respectively at 28 days of age, which was mainly due to less Ca and Al content that led to the formation of little C-(A)-S-H gels. This result was consistent with Rashad’s [45] work, wherein the CS of GPMs was found to decrease with the increase on fly ash content. This lowering in the mortars’ CS can be attributed to many factors: first, the dissimilarity in the chemical constituents of WCT, slag, and fly ash that appreciably influenced the binders’ alkali activation processes; second, the lower reaction rate of WCT and FA compared to GBFS, wherein GBFS was dissolved partially [46]; third, lower compactness and density of GPMs matrix at a higher level of WCT and FA incorporation; fourth, alkaline activator low molarity (2 M) in which the CS primarily depended on the calcium oxide concentration that replaced the low quantity of sodium oxide. Eventually, the generation of a high amount of N-A-S-H, C-S-H, and C-A-S-H gels led to the CS enhancement of the proposed GPMs.

### 2.2. SSBS Test

Figure 2 displays the WCT:GBFS:FA proportion-dependent SSBS of the studied GPMs at a curing age of 28 days. The GPM composite cylinder’s bond strength decreased from 4.47 MPa (control sample) to 4.13 MPa with the increase in WCT level from 0 to 50% as the FA replacement, respectively. The NC substrate and 30° slant shear load carrying capacity of GPMs at 28 days of curing age dropped from 4.17 to 2.77 MPa with the increase in WCT content from 50 to 70%, respectively. Mixes having higher levels of GBFS or containing GBFS and FA only (GPMs_1_) showed much higher SSBS that mix designed with 70% of WCT. This clearly suggested that GPMs consisting of high levels of GBFS could generate more C-(A)-S-H gels compared to those prepared with lower levels of GBFS. The observed findings were ascribed to the low silicate levels and non-reactive properties of GPMs. The bond strength of the ternary blended GPMs was significantly enhanced as GBFS was increasingly substituted by WCT and FA, thus making the mortars more reactive. In addition, the low CaO content and reduction in N,C-A-S-H gels formation [47] was responsible for the reduced bond strength of WCT-based GPMs. With the failure zone emerging outside the bond zone [48], the 30° SSBS result for the GPMs was the key finding in this study.

Figure 3 illustrates the XRD results of GPMs containing a high level of WCT as a slag and FA replacement. It was observed that the intensity of calcite peak at 29.8° was significantly influenced by WCT content. The intensity of the peaks at 29.8° due to dense gels was decreased as WCT and FA contents were increased, particularly for the specimens made with 70% of WCT as GBFS replacement. Likewise, the intensity of calcite peak at 29.8° for the specimen prepared with 30% of FA as GBFS replacement decreased compared to the specimen with 0% FA. The increment in aluminosilicate with increasing WCT and FA significantly affected the geopolymeraztion process and reduced the gels’ formulation. For specimens containing 70% of WCT, the intensity of the peak at 43.6° due to mullite was reduced as GBFS and FA contents were decreased in the GPM matrix. It is well known that Al_2_O_3_ is the main oxide beside mullite formulation, and thus a reduction in GBFS and FA contents in GPMs negatively affected the aluminum oxide level, leading to a reduction in the amount of mullite during the polymerization process. The intensity of the XRD peaks at 40.5 and 43° due to calcite were reduced with the increase in WCT level beyond 50%. Four significant XRD peaks observed at 21, 28, 38.5, and 51° were due to quartz and the weak peak at 34.1° corresponded to nepheline. With the rising level of WCT, the content of non-reacted silicate increased, which in turn lowered the formation of products related to the C-S-H, C-A-S-H, and N-A-S-H gels, thus lowering the SSBS of GPMs from 4.13 to 2.77 MPa at the curing age of 28 days.

Figure 4 shows the SEM results of GPMs prepared with various WCT:GBFS:FA contents at 28 days of age. GPMs designed with 50 and 70% of WCT (GPMs_2_ and GPMs_9_) were imaged using SEM. GPMs containing 50% of WCT revealed dense structure with a low quantity of non-reacted and partially reacted particles. Furthermore, GPM containing 70% of WCT showed a higher amount of non-reacted and partially reacted particles (Figure 4b) than the one designed with 50% of WCT (Figure 4a). A higher amount of highly porous and non-reacted silica with poor morphologies was formed with the increase in WCT contents. Consequently, the SSBS of GPMs was reduced to 2.77 MPa at 70% of WCTand 4.13 MPa at 50% of WCT.

### 2.3. Resistance to Sulfuric Acid Attack

Figure 5 shows the effect of H_2_SO_4_ attack on the SSBS of GPMs prepared with high-volume WCT content. After 12 months of exposure under 10% of H_2_SO_4_ solution, the bond strength of all specimens was reduced. The values of SSBS of GPMs were inversely proportional to the WCT content, wherein they were decreased from 24.8 to 19.2% with the increase of WCT level from 0 to 50% as the FA replacement. A similar trend was observed with the increasing level of WCT, wherein the bond strength of the GPMs decreased from 19.2 to 0.22% with the increase in WCT content from 50 to 70% as the GBFS replacement. Both led to an enhancement in the durability performance of the prepared specimens, lowering the loss SSBS percentage. In short, irrespective of the increasing WCT level (50, 60, and 70%), the replacement of GBFS by FA had a positive effect, which improved the durability of GPMs, presenting higher strength performance against the sulfuric acid attack.

The residual bond strength of the GPMs was enhanced when a high amount of WCT and FA in place of GBFS was incorporated in the geopolymer matrix. Additionally, the deterioration of the GPMs’ surface was restricted with the increase in WCT content (Figure 4). This observation was attributed to the formation of expansive ettringite and gypsum due to sulfuric acid attack. It may be accompanied by the expansion or softening of geopolymer mortars in which an increase in WCT level led to reducing the Ca^+2^ content within the geopolymer matrix with a lower CaSO_4_ product. Furthermore, the loss of bond strength with the increase in WCT to FA ratio in the geopolymer matrix occurred. The same observation was made when WCT and FA increasingly replaced GBFS, which reduced the calcium content (Table 1) and led to less internal deterioration because of lower CaSO_4_ formulation and expansion. The surface deterioration and crack formation were reduced with the increase in WCT content. Upon exposing the GPMs to acid, the calcium hydroxide compound present in GPM specimens could react with SO_4_^−2^ ion, forming gypsum (CaSO_4_·2H_2_O). This led to the expansion of the GPM matrix, and additional cracks were developed inside the specimens, as observed through visual appearance inspection (Figure 6). Many reports [49,50,51,52] revealed that due to the increase in SiO_2_, Al_2_O_3,_ and Na_2_O contents, the gypsum formation was reduced, thus increasing the durability of GPMs.

Figure 7 shows the XRD pattern of prepared GPMs after 360 days of exposure in 10% of H_2_SO_4_ solution. The GPMs samples displayed the presence of gypsum. From the analysis results, it was found that the calcite peak at 29.8° was replaced by gypsum after exposure to sulfuric acid for one year. Upon exposing the mortars in sulfuric acid, due to the dissolving of C-(A)-S-H gels and calcite and formation of more calcium hydroxide (Ca(OH)_2_) in the matrix, the Ca(OH)_2_ compound in the mortar reacted with the SO_4_^−2^ ion and formed the gypsum (CaSO_4_·2H_2_O). The most intense XRD peaks (at 26.8°, 40°, and 50°) corresponded to the crystalline phase of SiO_2_. The observed new peaks at 12.8° 16°, 20.7°, 22.4°, and 29.8° in the mortar prepared with various WCT were gypsum crystallites. The inclusion of WCT as a GBFS replacement could improve the GPMs’ durability and performance in acid environments (Figure 7a). The intensity of the gypsum peak at 29.8° decreased with the increase in WCT content. It is well known that the substitution of GBFS by WCT could control the CaO content, successfully controlling the formation of calcium hydroxide and CaSO_4_. H_2_O. Less formulation of CaSO_4_. H_2_O reduced the internal stress from gypsum expansion, resulting in a lower number of internal and external cracks and thus higher performance. Likewise, it was found that the replacement of GBFS by FA at the same level of WCT could positively affect the durability performance against acid attack, displaying lower deterioration. The weakening of the gypsum peak intensity at 29.8° was more pronounced with the increase in fly ash level (Figure 7b). Furthermore, the gypsum peak at 11.8° became more intense and closer to the SiO_2_ and CaSO_4_ peaks at 20.8° and 20.9°. As reported by Bellmann and Stark [53], the quartz and gypsum peaks can be difficult to distinguish. However, two peaks were probed in this study, corresponding to quartz and gypsum. An increase in the fly ash level to 30% as the replacement of slag restricted the CaSiO_4_ formulation, as can be clearly seen by the peak at 29.8°, which enhanced the GPMs’ durability performance in an aggressive environment (sulfuric acid attack). This suggested that the high content of GBFS in the GPMs matrix could lead to the formulation of more N, C-A-S-H gels that are vulnerable to sulfuric acid attack.

Figure 8 displays the effect of the H_2_SO_4_ solution environment on the surface morphology of GPMs. The performance of GPMs after immersion in 10% of H_2_SO_4_ solution for a period of 360 days was evaluated. The results indicated that, with increasing WCT content as GBFS replacement in the GPMs, the gypsum and internal cracks were reduced, lowering the deterioration. GPMs containing 50% of GBFS (Figure 8a) showed a higher amount of gypsum and ettringite with more cracks and porous morphology than those prepared with 70% WCT as GBFS replacement. It was demonstrated that [54] a reduction in the calcium content and controlled gypsum formulation in the GPMs and cement matrix was responsible for the enhancement in the binder performance when exposed to a sulfuric acid environment.

### 2.4. Resistance to Sulfate Attack

The loss in SSBS of the cylindrical GPMs–NC specimens containing a high volume of WCT was measured to assess the specimens’ performance in an MgSO_4_ environment (Figure 9). In general, all GPMs prepared with a high volume of WCT displayed excellent resistance compared to those containing higher GBFS. With the increase in WCT content in the geopolymer matrix from 50 to 70% as the replacement to GBFS, the SSBS dropped from 3.18 to 0.02%, respectively. In each level of high volume WCT geopolymer, the replaced GBFS by FA has positively affects residual slant shear bond strength, and loss bond strength decreased with increased FA content. When GBFS was replaced by WCT at 50, 60, and 70% in the geopolymer matrices, the resistance of the GPMs against sulfate environment was improved. GPMs prepared with a high level of WCT and FA as GBFS displayed very high resistance and showed a gain in the SSBS between 1.56 and 3.34%. In addition, the deterioration of the GPMs’ surface was lowered with the increase in WCT and FA contents. Most of the reports argued that sulfate attack could lead to the formation of expansive ettringite and gypsum, accompanied by the expansion or softening of geopolymer composite specimens. Consequently, with the decrease in GBFS content in the binder matrix, the Ca^+2^ content dropped with lower CaSO_4_ product, improving the GPMs’ resistance against sulfate attack. Additionally, the loss slant shear bond strength with decreased FA as GBFS replacement in a high-volume WCT matrix occurred because of same reason, as the GBFS replaced by FA decreased the content of calcium and led to more internal deterioration because of CaSO_4_ formulation and expansion. It was concluded that all GPMs–NC composite specimens displayed excellent performance in sulfate environments. However, no deterioration was observed on all surfaces of GPMs specimens containing a high amount of WCT and FA and exposed to sulfate attack.

### 2.5. Compatibility between Concrete Substrate and GPMs

Figure 10 illustrates the effects of various WCT:GBFS:FA contents on the de-bonding of the prepared GPM-NC composite blocks. The specimens prepared with high proportions of WCT showed high thermal compatibility, which remained constant after the 25 freezing–thawing cycles. As WCT levels in the GPMs were raised, the compatibility between the concrete substrate and the GPMs decreased, increasing the de-bonded percentage. Compared to the control sample (containing 50% of FA), the replacement of FA by 50% of WCT led to an increase in the de-bonded percentage (from 24 to 31.2%) of tested specimens. Likewise, the de-bonded percentage increased from 31.2 to 44.8% with the rise of WCT content from 50 to 70%, respectively. GPMs enclosing high amounts of WCT and GBFS replacement by FA showed increased de-bonded percentage. Furthermore, GPMs designed with 60% of WCT or lower showed stronger resistance to thermal expansion due to their higher level of compatibility compared to the one containing 70% of WCT. A lower amount of dense gels formulation could lead to the development of higher porosity in GPMs that were highly affected by the freezing–thawing, presenting lower resistance to ice expansion with high deterioration.

Figure 11 depicts the FS of the GPMs–NC composite beam as a function of varying WCT:GBFS:FA levels. It is known that the fragile materials deflect more in flexural tests compared to rigid materials. In the present GPMs, with the increase of WCT levels, the FS of the mortars decreased. Composite beams designed with 60% or lower WCT content showed higher FS than the composite NC (1.64 MPa). Additionally, with the increase in WCT levels from 50 to 70%, the FS of GPMs dropped from 1.73 to 1.58 MPa, respectively. When GBFS levels were below 50%, an increase in the WCT and FA contents affected the beam resistance and reduced the strength of GPMs to lower than that of the concrete substrate beam. The effects of WCT’s increasing levels on the failure mode of GPMs were assessed to ascertain the compatibility levels of the repair materials (Table 1). Type B failure mode occurred for the mortars that contained 50% of WCT or FA incorporated with GBFS. Additionally, the failure zone was shifted from B to C when the content of WCT and FA in the mortar was increased and the GBFS levels were changed from 30 to 40%. The composite beam designed with 70% of WCT produced Type D failure mode. The obtained high compatibility clearly indicated the benefit of the designed GPMs as a potential repair material, wherein the mortar made with 50% of WCT incorporated with GBFS revealed the optimum result.

Figure 12 demonstrates the flexural strength (FS) of GPMs for different values of WCT:GBFS:FA. Four-point load tests on prism specimens were carried out to measure the FS. GPMs containing more than 40% of GBFS revealed stronger resistance and bonding strength with NC compared to those mixes designed with lower than or equal to 30% GBFS. As WCT levels increased, the bond strength values of the GPMs decreased. In addition, it was observed that full replacement of FA by WCT (50%) could reduce the FS of the composite beam from 2.1 to 1.97 MPa. Likewise, the bond strength dropped from 1.97 to 1.73 MPa, when the WCT levels were raised from 50% to 70%, respectively. Conversely, GPMs cast with 50% of WCT as GBFS replacement by FA from 0 to 30% could increase the bond strength and reduce the FS from 1.97 to 1.73 MPa, respectively. A similar trend was observed for GPMs containing 60 and 70% of WCT wherein the bond strength became weaker with the increase in FA content as a GBFS replacement. The specimens cast with 50% of WCT or fly ash incorporated slag displayed failure zone type A (Figure 13). However, with the rise of WCT content up to 50% and reduction in slag to 20 and 30%, the failure zones were changed from Type A to Type B and C as opposed to Type C failure for OPC. Briefly, the variation of WCT:GBFS:FA value could significantly affect the failure zone, reducing the bond strength and compatibility with the increase in WCT and FA levels. Compared to the OPC beam (failure zone Type C that occurred closer to the bond zone on the concrete side), GPMs presented very high performance as the concrete RMs. This confirmed that the proposed GPMs, due to their extraordinary performance, can be nominated as potential surface repair materials.

Figure 14 presents the influence of varying WCT contents (at high volume) on the drying shrinkage (DS) of the ternary blend of geopolymer binders. The durability of concrete is its ability to perform satisfactorily when exposed to harsh environments over a prolonged time with minimum maintenance. The DS of a geopolymer at an early age is generally considered a critical parameter for the durability designing of the concrete structures. DS is a time-dependent deformation due to the loss of water by hydrostatic tension from the small capillary pores of the hydrated mortar specimens. Consequently, it may cause severe cracking in the concrete, allowing the ingress of aggressive agents inside the GPMs and eventual failure. Results showed that (Figure 14) the drying shrinkage values of the studied GPMs were significantly influenced by the inclusion of a high volume of WCT content in the geopolymer matrix. At an early age (3 days), due to the inclusion of WCT as full replacement for FA, the drying shrinkage was dropped from 208 to 192 microstrains (Figure 14a). Likewise, with the increase of WCT level from 50 to 70% the total shrinkage dropped from 192 to 166 microstrains, respectively. A similar trend was observed in each level of WCT, wherein the drying shrinkage value was decreased with the increase in FA as a replacement for GBFS.

At 28 days (Figure 14b) and 180 days (Figure 14c) of curing age, the drying shrinkage of GPMs displayed a similar decreasing trend with the increase of WCT content as a GBFS replacement. All the designed GPMs specimens containing WCT showed lower drying shrinkage than the one obtained for OPC-based NC. Previous studies indicated that the drying shrinkage characteristics of the alkali-activated geopolymer composites are more complicated than OPC composites due to their complex hydration and shrinkage mechanisms [55,56,57]. Among these geopolymer binders, the shrinkage problem of GBFS-based geopolymer systems is particularly prominent, which is significantly higher compared to other geopolymer composites (such as FA, metakaolin, and coal gangue) [56]. Some reports indicated that the partial replacement of GBFS by FA can effectively alleviate the shrinkage of the GPMs systems but weaken their strength development [22,32,54,55]. Therefore, the selection of suitable proportions of the mineral admixtures is very important to optimize the performance of GPMs. It was shown that the inclusion of FA can enable the formation of some 3D networked N-A-S-H gels, restraining the shrinkage of C-A-S-H gels [58]. According to Humad et al. [59], the addition of FA in the AAS mixture can significantly mitigate its autogenous shrinkage without exhibiting considerable effects on its drying shrinkage. So far, only a few studies have suggested that the incorporation of FA in the geopolymer matrix can enhance the shrinkage of GPMs [55]. The effects of FA addition on the drying shrinkage were attributed mainly to the increased porosity and reduced tortuosity of GPMs [58,60,61].

## 3. Conclusions

New types of GPMs using various industrial wastes (WCT, GBFS, and FA) were designed and characterized to determine their potency as high-performance repair materials. To meet this target, ten GPMs and control OPC concrete mixtures were made and tested. The SSBS performance of the composite GPMs–NC (cylindrical specimens) under the aggressive environments was evaluated. In addition, the mechanical properties of GPMs were measured using the CS, SSBS, and FS four-point loading and thermal expansion coefficient tests. The compatibility between the base concrete substrate and designed GPMs was evaluated. It was shown that the proposed GPMs can effectively be used as the repair materials for the damaged concrete surfaces under aggressive environmental conditions. Based on the obtained results, the following conclusions can be drawn:i.Sustainable geopolymers as concrete repair materials were produced incorporating WCT, GBFS, and FA. The content of calcium in the design mix was significant where the main recourse of calcium (GBFS) was kept between 20 and 70%.ii.Both the CS and SSBS of GPMs were decreased with the increase in WCT and FA content as GBFS replacements. At 28 days of age, GPMs made with 70% WCT and 20% GBFS showed lower bond strength than the control specimen. The XRD and SEM analyses showed that with the increase of WCT and FA content as GBFS replacements, GPMs became more porous and less homogeneous with lower amounts of N-A-S-H and C-A-S-H dense gels formation.iii.With the increase in WCT and fly ash as slag replacement, the performance of the composite specimens was enhanced and the loss in SSBS was reduced when exposed to sulfuric acid environments. Composite GPMs–NC (cylindrical specimens) prepared with 70% of WCT displayed excellent performance and achieved the highest resistance against sulfuric acid and sulfate attacks. It was claimed that the GPMs made with a high volume of WCT can be highly recommended as the sustainable repair materials for aggressive environments.iv.XRD and SEM microstructures analyses of GPMs exhibited that, with the increase in WCT level as a GBFS replacement, the formation of gypsum and ettringite in the geopolymer matrix were restricted, which in turn reduced the internal cracks and deterioration of the proposed repairing materials.v.The drying shrinkage of GPMs was reduced due to the inclusion of WCT as a GBFS replacement. All the specimens prepared with WCT as a GBFS replacement showed lower drying shrinkage compared to OPC-based NC. This reduction in the drying shrinkage could positively affect (increase) the durability performance (lower deterioration) of GPMs and GPMs–NC composites, thus increasing their life span because of the formation of fewer cracks in the mortars matrix.vi.The obtained GPMs showed a high degree of suitability for repair works in terms of very excellent GPM–concrete substrate compatibility levels, as shown by the SSBS, CTE, and FS of four-point loading GPMs–NC composite beam results.vii.Based on the CS and SSBS resistance against aggressive environmental conditions (sulfate and sulfuric acid exposure), thermal expansion coefficient, four-point flexural strength of composite beam, and drying shrinkage results, GPMs composed of 50% WCT, 40% GBFS, and 10% FA can be highly recommended as a potential concrete repair material in the construction sector worldwide.

## 4. Materials and Methods

### 4.1. Materials Characterization

In this work, all the raw binder materials (WCT, GBFS, and FA) were procured from one source and kept in airtight plastic vessels to prevent contamination to develop the GPMs. The obtained WCT was ground prior to its use. Table 2 shows various physical attributes of the binder components. To produce the binder free of cement, GBFS and FA (with no additional purification) were used as the constituent. Because the GBFS has cementitious and pozzolanic properties, the powder is distinct from other auxiliary cementitious substances (it is off-white in color). Once the water had been mixed, the GBFS came out of the hydraulic chemical reaction. FA with low calcium content (source of aluminosilicates) was collected from the power station to make the GPMs. Table 2 displays the chemical constituents present in WCT, GBFS, and FA determined using the XRF test. Silicon and Al were the major elements with approximately 84.8%, 41.7%, and 86% in WCT, GBFS, and FA, respectively. WCT was made up of about 72.6% silica, GBFS was made up of around 51.8% calcium oxide, and FA was made up of about 28.8% aluminum oxide. It is known that the presence of Si, calcium oxide and Al play a vital role in the GPMs to form N-A-S-H and C-(A)-S-H gels during the geopolymerization. WCT contained a higher amount of Na_2_O (13.5%) than slag (0.46%) and fly ash (0.07%). Very low loss on ignition (LOI) content was evidenced in WCT, GBFS, and FA. The GPMs made from FA displayed the lowest amount of C-A-S components, consisting of 5.2% Ca, 57.2% silicate, and 28.8% Al, which was grey in color, thus making it suitable for classification as FA (class F) according to the ASTM C618. The medians of the particles for WCT, GBFS, and FA (measured using particle size analyzers) were 35, 12.8, and 10 µm, respectively. To evaluate the physical properties of materials, the Brunauer-Emmett-Teller (BET) test was performed. From the results, fly ash exhibited a greater surface area (18.1 m^2^/g) compared to both WCT (12.2 m^2^/g) and slag (13.6 m^2^/g).

Figure 15 illustrates the X-ray diffraction (XRD) profiles of WCT, GBFS, and FA powder, which consisted of intense peaks at 2θ rage of 16–30°. These peaks were allocated to the existence of silica and alumina crystallites. Occurrences of other sharp peaks were assigned to the existence of quartz and mullite crystallites. Many reports [62,63,64] claimed that the glassy nature of WCT and FA are vital for the hydration reaction and formation of gels. The absence of a sharp XRD peak of GBFS indeed confirmed their true disordered nature. The occurrences of silica and calcium peaks in the XRD pattern were significant towards GBFS creation. The presence of high content of reacting amorphous silica and calcium oxides in slag was advantageous for GPMs synthesis. However, the inclusion of fly ash was essential to surmount the low level of Al_2_O_3_ (10.49 wt.%) in slag.

To make the new mortar types, washed sand (siliceous) drawn from a river was used to make up the fine aggregates. In accordance with ASTM C117 [65], the study first cleaned the sand to reduce the levels of silts and impurities. Then, it was oven dried (at 65 °C) to eliminate all moisture prior to grading it in accordance with ASTM C33–33M [66]. The specific gravity and fineness modulus of the natural aggregates were 2.6 and 2.9. To produce the solution with a concentration of two molarity (7.4% of Na_2_O and 92.6% of H_2_O), analytical-grade sodium hydroxide (NH; 98% of purity) was dissolved into water as pellets. An analytical grade sodium silicate (NS) blend with H_2_O (55.80 wt.%), Na_2_O (14.70 wt.%), and SiO_2_ (29.5 wt.%) was developed for the study. The solution was stored for 24 h to bring the temperature to room temperature. To produce the alkaline mixture with a 1.2 ratio of SiO_2_:Na_2_O in which the proportion of sodium silicate to sodium hydroxide for the all-alkaline mixes was maintained at 0.75, this was mixed with the sodium silicate solution.

### 4.2. Design of GPMs Mixes

This section describes the effect of WCT and FA replacing GBFS at various levels on the contents of SiO_2_, CaO, and Al_2_O_3_ in GPMs. Three levels of replacement were adopted to evaluate the effect of WCT content on the geopolymerization process and durability performance of proposed mortars. In each level, the minimum content of GBFS was kept as low as 20%, as presented in Table 3. Furthermore, at every level, the GBFS was replaced by FA to evaluate the effect of increasing silica oxide and aluminium oxide on GPM properties. The GPMs as repair materials were comprised of the binder (B) to fine aggregate (A) (B:A) and alkaline activator solution (S) to binder (B) (S:B) ratio of 1.0 and 0.40, respectively. As no standard was clean cut, the values of B:A and S:B across the GPM mixtures when deployed as repair materials stood at 1.0 and 0.40. Observations of the trial mixtures that realized the highest compressive strength at a curing age of 28 days were used to determine the value of B:A because no standardized guidance was available in the preparation of GPMs [67,68]. The GPM control mix designs were prepared using two forms of industrial waste (GBFS and FA). The ratio of 50% of GBFS to FA was chosen as the control batch, and then the GBFS and FA were replaced by a different level of WCT. These were blended to identify the effects of CaO, Al_2_O_3_, and SiO_2_ content on the process of geopolymerization under the acid environment. The GBFS content was maintained in a range of 50 to 20% as the source of CaO. Sodium hydroxide (NH) molarity, sodium silicate (NS) to NH ratio, and the modulus (Ms) of the alkaline solution (SiO_2_ to Na_2_O ratio remained constant across all the mixes.

During the GPMs mixing, total river sand was placed in the mixer, followed by the addition of 50% of binary/ternary blended (FA, GBFS, and WCT). The materials were mixed in dry conditions for three minutes to create a homogenous mixture before adding the second batch of binder (50%). After including the second batch of binder, the materials were mixed for another two minutes with normal speed. Finally, the mixed materials (river sand and binders) were activated with the prepared alkaline activator solution (sodium silicate and sodium hydroxide). After a further three minutes of mixing, the final mixture was poured in the molds. The casted specimens were left for one-day curing at room temperature (26 °C) and relative humidity of 75%. Later, the cured GPM specimens were de-molded and left in the same condition of curing until the day of test. For the concrete substrates (NC), the ordinary Portland cement (OPC) was used to manufacture the NC with compressive strength up to 40 MPa. In the trial mixture, the OPC to river sand to crushed stone proportion was 1:1.5:3.0 and 0.48 was the water to OPC ratio. The strength properties of NC specimens were evaluated at 28 days of curing. The tested specimens displayed 43.61 MPa compressive strength (CS), 5.61 MPa flexural strength (FS), 4.43 MPa splitting tensile strength (STS), and 28.21 GPa modulus of elasticity (MoE).

### 4.3. Compressive and Bond Strength Tests

Following to ASTM C109 [69], the compressive strength of GPMs specimens with various ratios of WCT:GBFS:FA were evaluated at 28 days of curing. The average value of the tested three specimens was considered for each mixture. For all the GPM specimens (50 mm × 50 mm × 50 mm), the consistent loading rate (2.5 kN/second) was applied using a compression strength machine. For the slant share bond strength (SSBS) test of the hardened NC, it was diagonally positioned at a 30° inclination from the vertical, and the bond strength between GPMs and the concrete substrate was measured. A closer minimum stress with bond angle of 30° constituted the failure stress point accordant with a smooth surface following to ASTM C882. The typical procedure of the SSBS test was presented in Figure 16. The NC specimens were created utilizing the aged shear concrete cylinders at the 30° inclination. The SSBS evaluation was performed on the saw cut concrete surface. The conventional concrete was placed half-slanted into the cylinder mold, and the fresh GPMs were filled in the cylinder mold (Figure 17). The CS test was carried out after 28 days of curing where a compression machine was used to conduct this test.

### 4.4. Resistance to Sulfuric Acid Attack

Sulfuric acid (H_2_SO_4_) mainly affected the GPMs by weakening and disintegrating the binder paste solution. Deionized water (DIW) was utilized to prepare a 10% concentration of the H_2_SO_4_ solution and its effects on the GPMs’ composition were analyzed. For a total of six GPM specimens, each 28days old, the weight was measured and recorded before immersion the composite cylinders (GPMs-NC) into acid solution for one year. In order to maintain a fixed pH level, the sulfuric acid solution was changed every three months. All GPM-NC cylinders were checked after one year where different factors were considered during the performance assessment, such as the deterioration degree of microstructure and the loss in weight and residual SSBS based on ASTM C267 stipulation [70]. The (SO_4_)^2-^ ions at different concentrations in water are the main reason for the acid attack of GPMs, which were transferred to the mortar along with Mg, Ca, or Na cations.

### 4.5. Resistance to Sulfate Attack

Sulfate attack of GPMs was essentially caused by the presence of (SO_4_)^−2^ ions in water at different concentrations that were transported into the mortar along with Ca, Mg, or Na cations. MgSO_4_ solution (QREC, Malaysia) was utilized to evaluate the durability performance of GPMs against sulfate attack. At 28 days of curing age, the specimens are weighted to obtain the initial weight, and they are immersed in 10% magnesium sulfate solution for a duration of one year. The solution waste was removed in every three months to maintain a constant pH level. The residual weight and strength were measured based on ASTM C267 (2012) specification.

### 4.6. Compatibility between GPM and NC Substrate

In this study, several tests were adopted to evaluate the compatibility between the GPMs as repair material and NC, which includes coefficient thermal expansion (CTE), the flexural strength (FS) test for four-point loading and the drying shrinkage (DS) test. The CTE is a measure of the length change in composite GPMs–NC specimens exposed to cycles of freeze-thaw (change in temperature). When composite materials (RM–NC) of different CTE are joined together and subjected to significant temperature changes, stresses are generated in the composite specimens. The compatibility between concrete substrate and GPMs was evaluated using a thermal expansion coefficient test. By exposing them to temperature variations and measuring the alterations in length, the study gauged the coefficients of thermal expansion (CTEs) of the GPMs. Stresses were generated once the GPMs were connected and the substrate composite of dissimilar CTE was combined and exposed to temperature variations. Failures at the composite’s interface or in the materials of less strength were caused by the ensuing stress. To avoid failure at the higher fluctuating temperature, the GPMs must have similar CTE to the NC. CTE is a vital factor for the repairing component when exposed to varying temperatures. The ASTM C884 was altered (it is ostensibly designed as a thermal compatibility test between an epoxy resin overlay and concrete) to develop a thermal consistency test for the GPMs. A base for the tests was formed by an 80 mm × 100 mm × 200 mm concrete block (Figure 18). Such blocks should sustain 10 freezing-thawing cycles, as in ASTM C666 (which was originally designed for assessing concrete resistance in rapid freezing-thawing cycles or Method A). The substrate concrete blocks were subjected to a range of repairing mortars, with the thickness of the blocks ranging 10 to 12 mm after 28 days of curing. Thereafter, the blocks were exposed to 25 cycles of freeze–thaw, with the varying cycle temperatures ranging from 5 to 20 °C. A qualitative test of de-bonding and visual inspection were performed at the closure of each cycle in order to identify signs of cracks, scaling, or bind breakage [71].

Two key procedures were applied to test the concrete substrate and compatibility between the GPMs. In the first test (ASTM C78 standard), GPMs were poured to the depression generated at the bottom of a prism-shaped concrete piece (dimensions 250 mm × 100 mm × 10 mm) as shown in Figure 19a. At 28 days of curing, the concrete substrate showed the strength of 43.6 MPa. The filled side of the GPMs was placed at the specimens’ bottom in this test. In test 2, the vertical shear bonding strength (SBS) between GPMs and NC (dimensions 100 mm × 100 mm × 30 mm) was determined (Figure 20a). Prism shaped specimens with 40 MPa grade (C40) were cast (dimension of 100 mm × 100 mm × 500 mm). First the specimens were cut at the middle to the required dimension and then put into the prism molds to pour the fresh GPMs. A four-point load (at a rate of 0.2 kN/s) was then deployed to test the samples after 28 days of curing. The failure mode was used to evaluate the compatibility of the repair materials and the concrete substrate. Once the failure passed through the concrete substrate and repair material at the middle third of the prism, the GPMs were likely to have adequate compatibility. When the GPMs did not meet this requisite, they were declared incompatible. Figure 19b and Figure 20b display the failure zone types at the bonding stresses of the composite beam and the four-point loading. The OPC mortar was used as a comparator (cement to sand ratio of 1:3, water: cement ratio of 0.48 and CS of 34.1 MPa after 28 days of curing). The compatibility between the NC and GPMs was measured in terms of the TCE, bending stress, and four-point loading flexural strength.

The DS test was conducted based on the procedure outlined in ASTM C157/C157M. For each GPM mixture, three sets of specimens (25 mm × 25 mm × 250 mm) in the form of prisms were adopted to measure DS at early and late curing ages. The specimens were prepared following the ASTM C192/192M and cured in the ambient condition. Stainless steel studs were fixed on the specimens to measure the change in length at 3, 28, and 180 days. The specimens were de-molded one day after casting and then moved to a chamber maintained at a temperature of (23 ± 1.5) °C and relative humidity of (50 ± 5)%. Subsequently, readings were taken using a DEMEC meter at the mentioned ages. The length change *(*Δlx) of the specimen at any age (%) was calculated via
(1)$$\Delta lx=\frac{CRD-initial\;CRD}{G}\times 100$$
where *CRD* is the difference between comparator reading of the specimen and reference bar at any age and *G* is the gauge length (100 mm).

## Figures and Tables

**Figure 1 gels-08-00053-f001:**
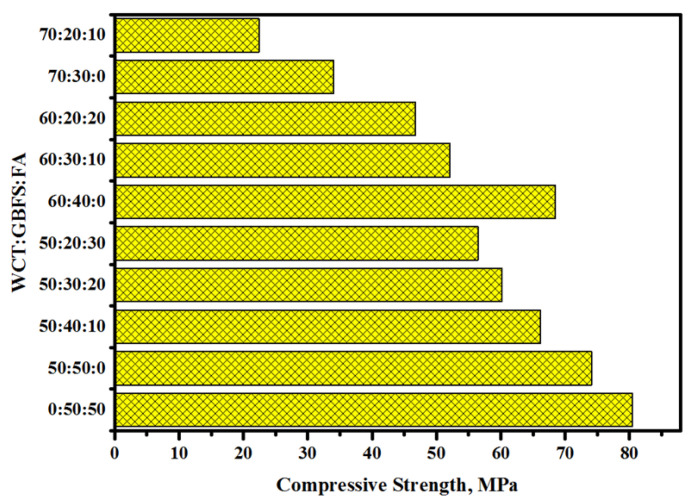
CS development in GPMs incorporated with WCT, GBFS, and FA.

**Figure 2 gels-08-00053-f002:**
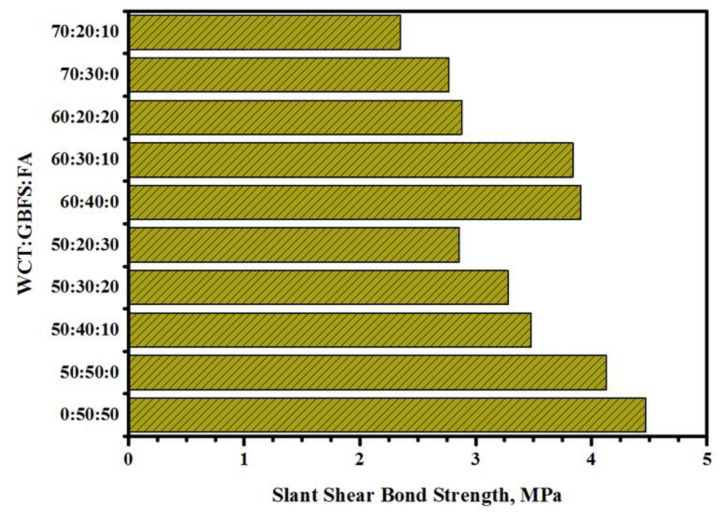
Effects of various WCT:GBFS:FA on SSBS of GPMs at 28 days of age.

**Figure 3 gels-08-00053-f003:**
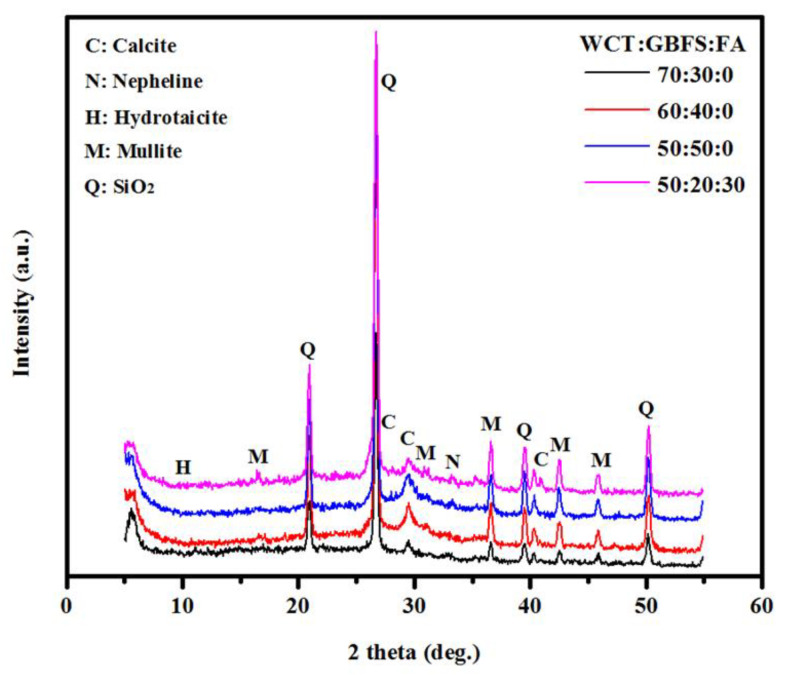
Effects of various WCT:GBFS:FA on the XRD patterns of GPMs.

**Figure 4 gels-08-00053-f004:**
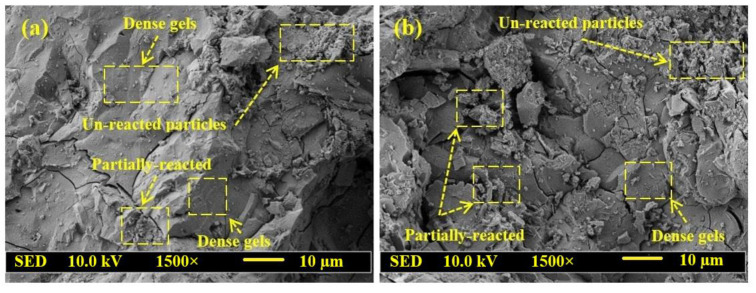
SEM images of GPMs at 28 days of curing age containing (**a**) 50% WCT and (**b**) 70% WCT.

**Figure 5 gels-08-00053-f005:**
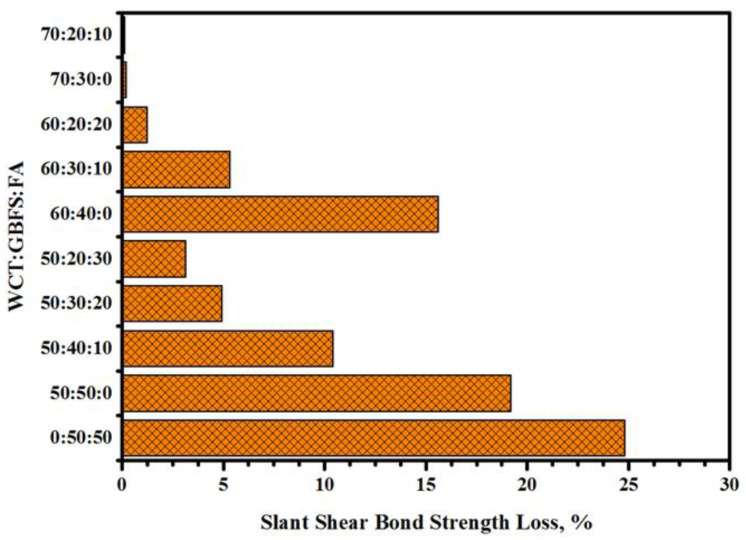
Effect of H_2_SO_4_ solution on SSBS performance of GPMs.

**Figure 6 gels-08-00053-f006:**
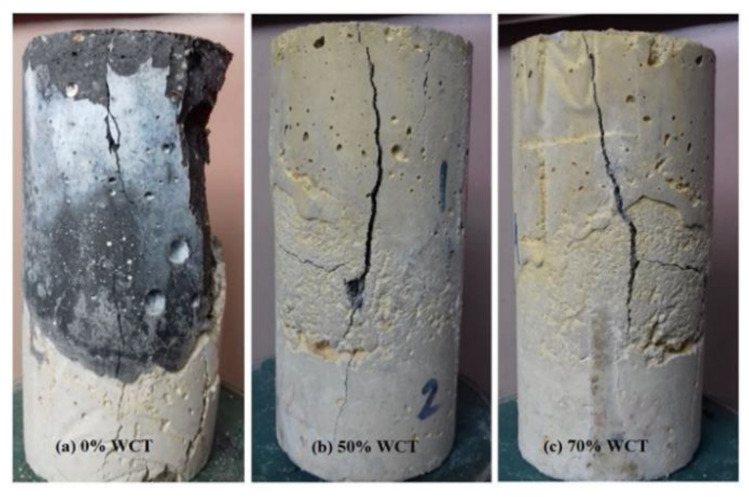
Failure mode of GPM specimens under 10% of H_2_SO_4_ exposure.

**Figure 7 gels-08-00053-f007:**
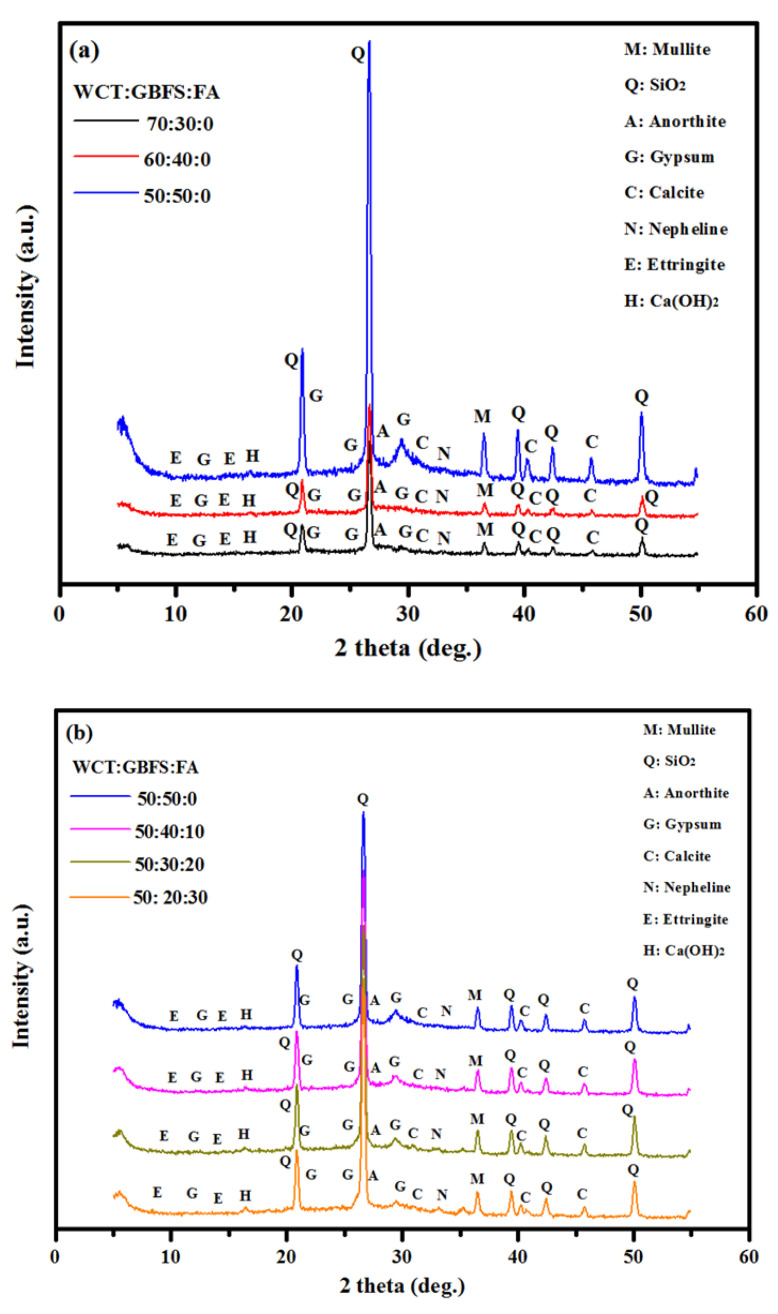
XRD patterns of GPMs exposed to sulfuric acid for 365 days (**a**) effect of various WCT levels and (**b**) effects of various FA levels.

**Figure 8 gels-08-00053-f008:**
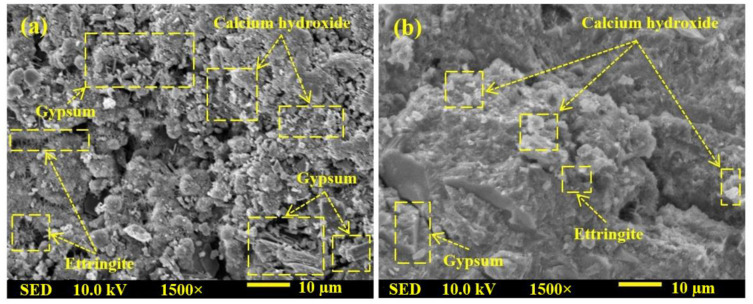
SEM images of GPMs exposed to 10% of H_2_SO_4_ for 365 days containing (**a**) 50% and (**b**) 70% of WCT.

**Figure 9 gels-08-00053-f009:**
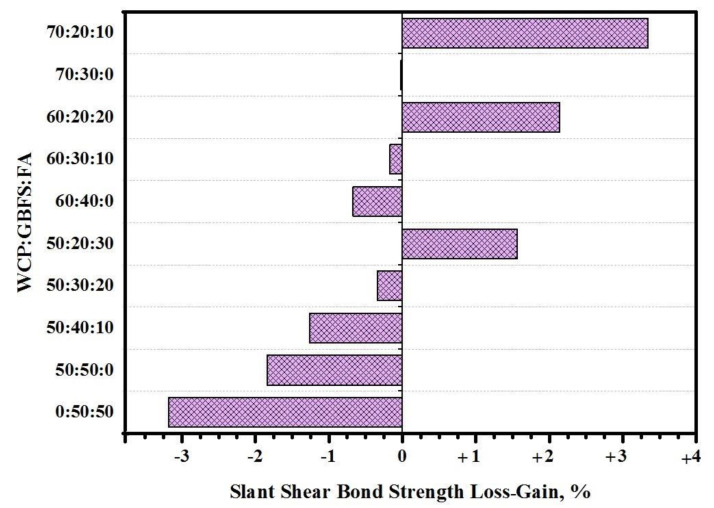
Effect of sulfate attack on the SSBS of WCT-based GPMs.

**Figure 10 gels-08-00053-f010:**
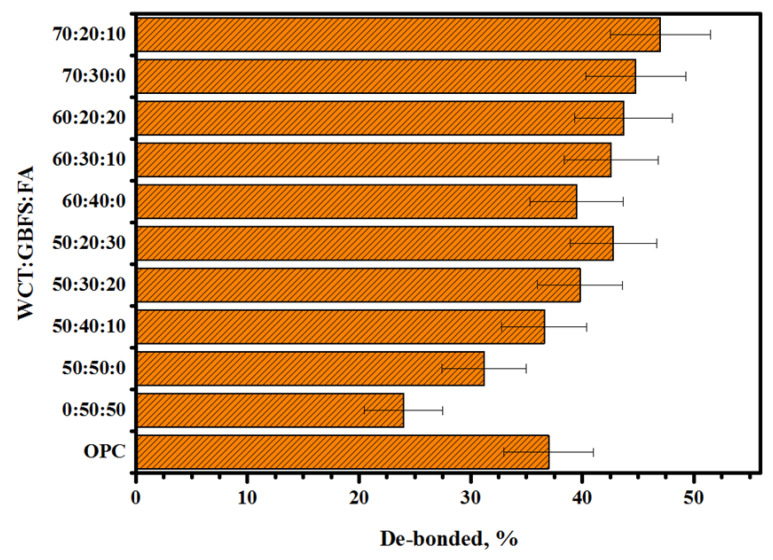
Effects of various WCT:GBFS:FA on the de-bonded percentage of the prepared GPMs.

**Figure 11 gels-08-00053-f011:**
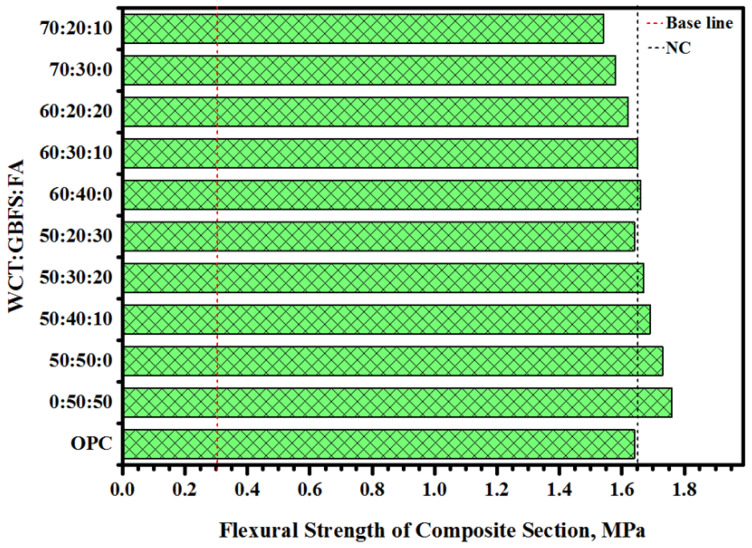
FS of composite beam prepared with various ratios of WCT:GBFS:FA.

**Figure 12 gels-08-00053-f012:**
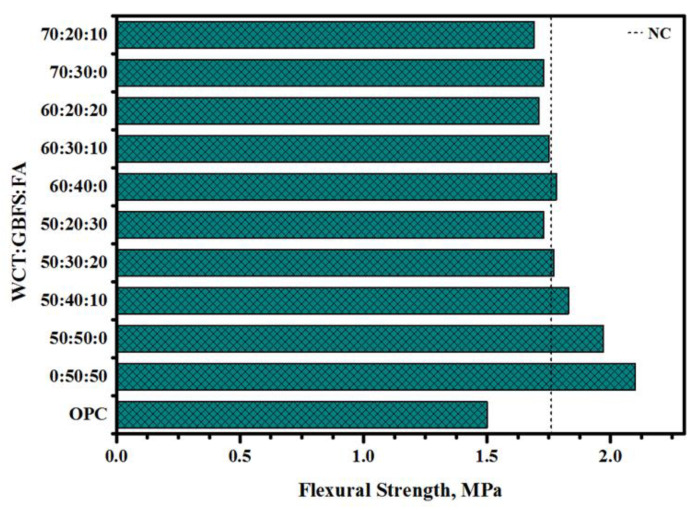
Impact of various WCT:GBFS:FA on the FS of four points of the repairing prisms.

**Figure 13 gels-08-00053-f013:**
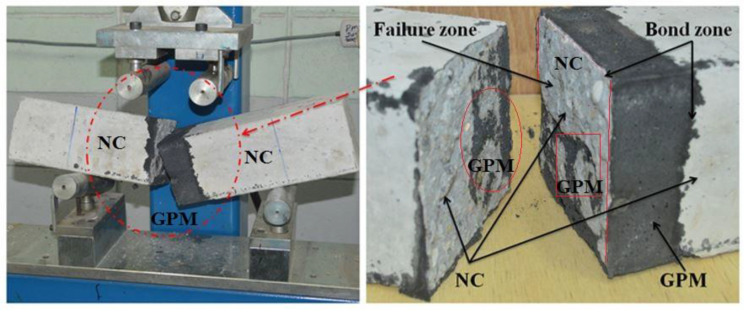
Failure pattern of GPMs made with 50% WCT under the four points flexural strength test.

**Figure 14 gels-08-00053-f014:**
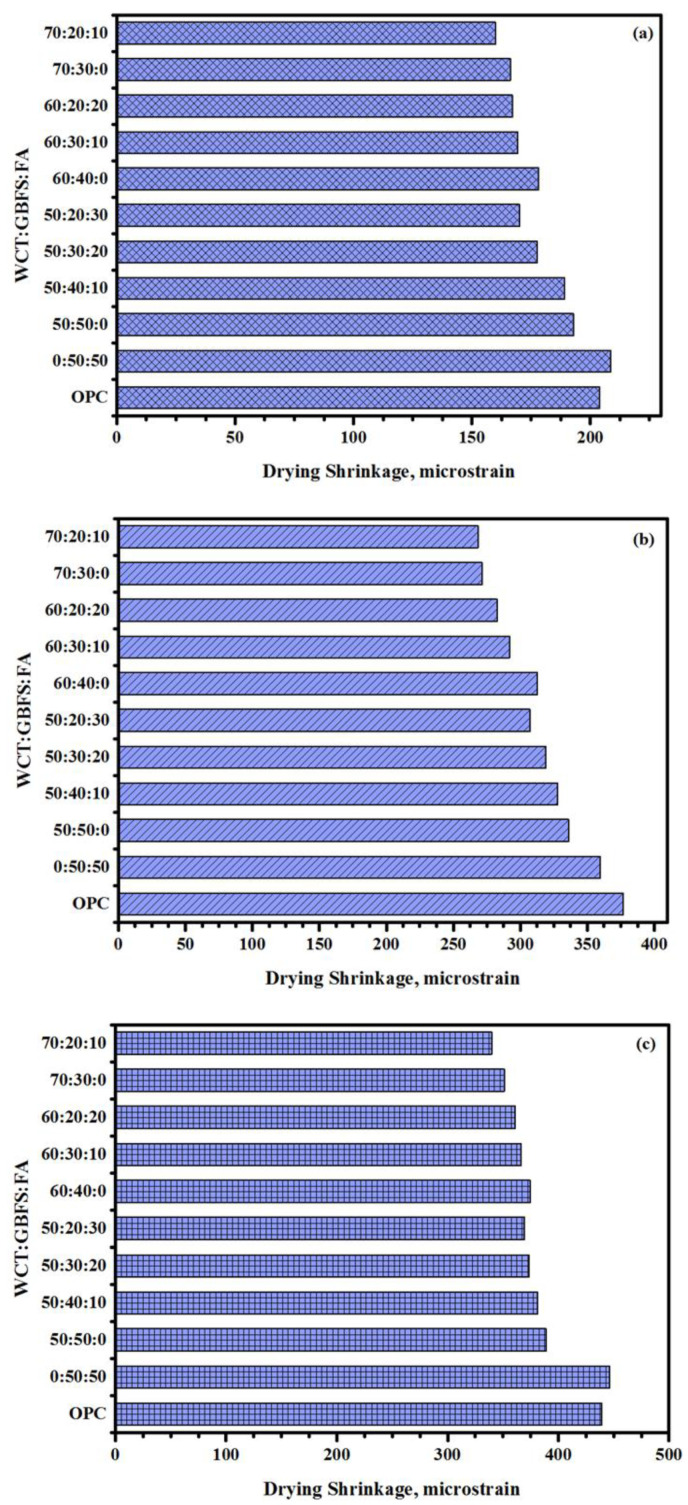
Effects of various WCT content levels on the drying shrinkage of GPMs at early and late ages (**a**) 3 days, (**b**) 28 days, and (**c**) 180 days.

**Figure 15 gels-08-00053-f015:**
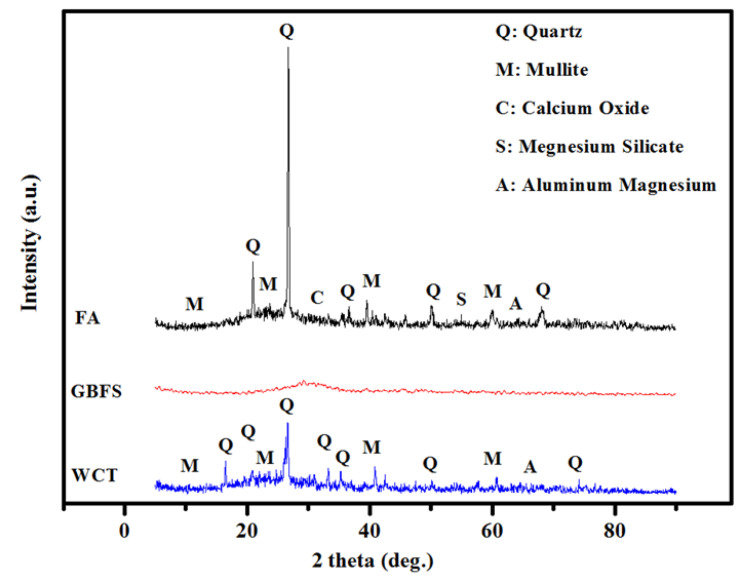
POFA lab treatment.

**Figure 16 gels-08-00053-f016:**
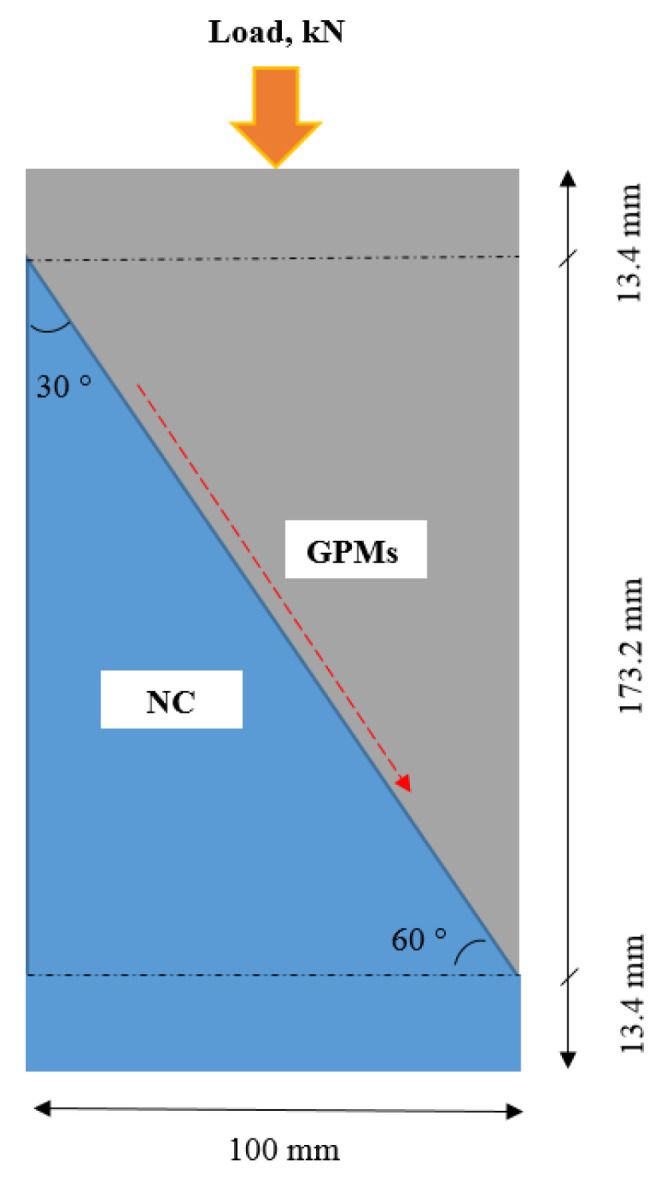
Geometry of SSBS test.

**Figure 17 gels-08-00053-f017:**
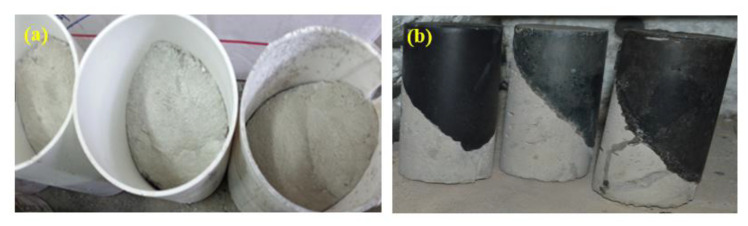
The steps to prepare the slant shear GPMs samples (**a**) concrete substrate (**b**) composite GPMs-NC.

**Figure 18 gels-08-00053-f018:**
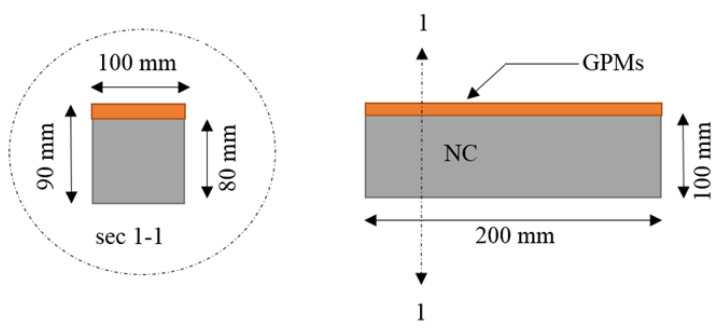
Compatibility between concrete substrate and GPMs.

**Figure 19 gels-08-00053-f019:**
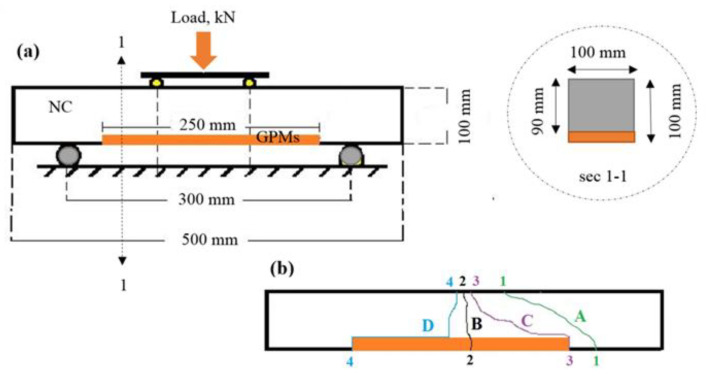
Compatibility evaluation according to failure mode: A, B-compatibility; C, D-incompatibility (**a**) composite beam and (**b**) failure zones.

**Figure 20 gels-08-00053-f020:**
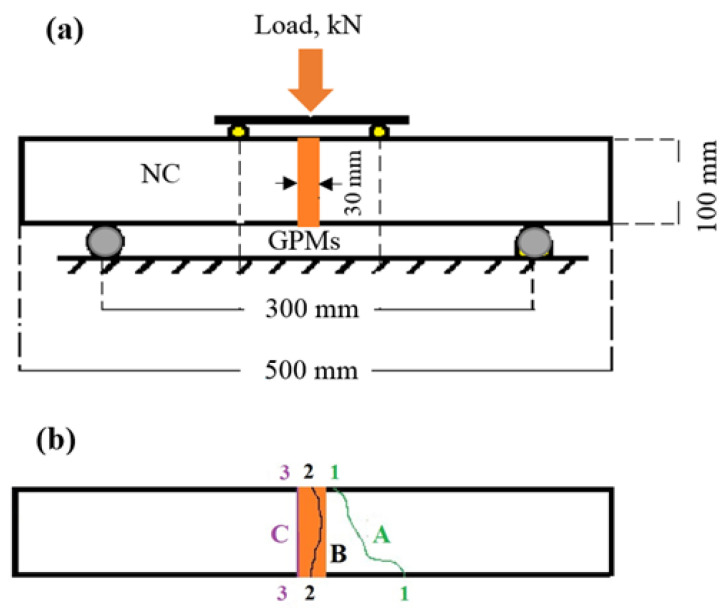
Compatibility evaluation according to failure mode: A, B-compatibility; C-incompatibility (**a**) GPMs–NC composite beam and (**b**) mode of failure.

**Table 1 gels-08-00053-t001:** GPMs–NC composite beam FS and failure zone.

Mix	FS (MPa)			Failure Zone
	Base Line (NC)	NC	GPMs-NC Beam	
OPC	0.63	1.67	1.64	D
GPMs_1_	0.63	1.67	1.76	B
GPMs_2_	0.63	1.67	1.73	B
GPMs_3_	0.63	1.67	1.69	B
GPMs_4_	0.63	1.67	1.67	C
GPMs_5_	0.63	1.67	1.64	D
GPMs_6_	0.63	1.67	1.66	C
GPMs_7_	0.63	1.67	1.65	D
GPMs_8_	0.63	1.67	1.62	D
GPMs_9_	0.63	1.67	1.58	D
GPMs_10_	0.63	1.67	1.54	D

**Table 2 gels-08-00053-t002:** Chemical and physical attributes of raw materials based on XRF test.

Chemical Composition			
Oxides/Material	WCT	FA	GBFS
SiO_2_	72.6	57.20	30.8
Al_2_O_3_	12.2	28.81	10.9
Fe_2_O_3_	0.56	3.67	0.64
CaO	0.02	5.16	51.8
MgO	0.99	1.48	4.57
K_2_O	0.03	0.94	0.36
Na_2_O	13.46	0.07	0.46
SO_3_	0.01	0.10	0.06
LOI	0.13	0.12	0.22
**Materials’ physical traits**			
Specific gravity	2.61	2.2	2.9
Surface area-BET (m^2^/g)	12.2	18.1	13.6
Medium diameter (µm)	35	10	12.8

**Table 3 gels-08-00053-t003:** Proposed GPMs mix design with various ratios of WCT:GBFS:FA.

Materials (Mass%)			Geopolymer Mortars of the Designed Mixtures									
			GPMs_1_	GPMs_2_	GPMs_3_	GPMs_4_	GPMs_5_	GPMs_6_	GPMs_7_	GPMs_8_	GPMs_9_	GPMs_10_
Binder (B)	WCT		0	50	50	50	50	60	60	60	70	70
	GBFS		50	50	40	30	20	40	30	20	30	20
	FA		50	0	10	20	30	0	10	20	0	10
	SiO_2_:Al_2_O_3_		2.82	4.48	4.08	3.77	3.53	4.79	4.35	4.01	5.09	4.62
	CaO:SiO_2_		1.68	0.50	0.39	0.29	0.20	0.37	0.27	0.18	0.26	0.17
	CaO:Al_2_O_3_		4.75	2.24	1.59	1.09	0.70	1.77	1.19	0.74	1.31	0.79
S:B			0.40	0.40	0.40	0.40	0.40	0.40	0.40	0.40	0.40	0.40
B:A			1.0	1.0	1.0	1.0	1.0	1.0	1.0	1.0	1.0	1.0
Na_2_SiO_3_:NaOH (NS:NH)			0.75	0.75	0.75	0.75	0.75	0.75	0.75	0.75	0.75	0.75
NaOH (NH)		H_2_O	92.6	92.6	92.6	92.6	92.6	92.6	92.6	92.6	92.6	92.6
		Molarity, M	2.0	2.0	2.0	2.0	2.0	2.0	2.0	2.0	2.0	2.0
		Na_2_O	7.4	7.4	7.4	7.4	7.4	7.4	7.4	7.4	7.4	7.4
Na_2_SiO_3_ (NS)		Na_2_O	14.7	14.7	14.7	14.7	14.7	14.7	14.7	14.7	14.7	14.7
		SiO_2_	29.5	29.5	29.5	29.5	29.5	29.5	29.5	29.5	29.5	29.5
		H_2_O	55.8	55.8	55.8	55.8	55.8	55.8	55.8	55.8	55.8	55.8
Total H_2_O in alkaline solution			76.8	76.8	76.8	76.8	76.8	76.8	76.8	76.8	76.8	76.8
Ratio of SiO_2_:Na_2_O (Ms)			1.2	1.2	1.2	1.2	1.2	1.2	1.2	1.2	1.2	1.2

## Data Availability

Data is contained within the article.

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
