# Peer review of "Effects of Sulfate and Sulfuric Acid on Efficiency of Geopolymers as Concrete Repair Materials"

_gels, 2022, doi:10.3390/gels8010053_

Round 1
Reviewer 1 Report
This paper represents the effect of using sulphate and sulphuric acid material to repair the concrete material. The paper is well written and appropriately indicated the methodology and results and discussion. However, I recommend authors to add and revise the following items:
1- The novelty should be added in the abstract.
2- Introduction needs to be rechecked. Some references have been added in introduction section but they are not explained in details. Authors should do the literature by mentioning the importance of the reported paper and explain their methods and findings.
3- In Figure 9 and 13, the maximum point of each category needs to be compared and the explanation should be added.
this paper can be accepted if the aforementioned items are revised.
Author Response
As attached

Reviewer 2 Report
Manuscript title: Effects of sulphate and sulphuric acid on efficiency of geopolymer as concrete repair materials
Manuscript ID: gels-1536771
The manuscript is interesting and well organized, but it needs some modifications before it can be accepted for publication, as follows:
- In the abstract, please use the passive voice instead of the pronoun "we".
- Please briefly mention the novelty of your work in the abstract.
- The novelty of the research is not well clear, please rewrite the novelty (in the last paragraph of the introduction), indicating the difference between your work and the literature.
- Please rewrite and shorten the conclusions section focusing on the main results. It seems more like a discussion than a conclusion.
Author Response
As attached

Reviewer 3 Report
1.General Comments:
In this paper, the purpose of this study is to analysis the effects of sulphate and sulphuric acid on efficiency of geopolymer as concrete repair materials. On the whole, this article is very good, the overall language of the article is logical and normative. This paper mainly carries out relevant tests on Geopolymer Materials under different mix proportions, including compressive strength, bonding strength, resistance to sulfuric acid and sulfate corrosion, as well as compatibility test between Geopolymer Materials and concrete substrate, which verifies the effectiveness of Geopolymer Materials used in this paper as repair materials. But before the article is published, there are some problems should be solved. The problems are as follows.
2.Problems:
Comment 1: Line 159: The determination of the binder (B) to fine aggregate (A) (B:A) and alkaline activator solution (S) to binder (B) (S:B) values can be described in the introduction, including the test results after using these values in the existing literature.
Comment 2: Line 181: In this paper, the curing time is at room temperature (26 ℃), and the relevant properties are measured after curing to 28 days. Whether it is necessary to supplement the early strength test of the specimen here needs to be considered due to the characteristics of Geopolymer Materials.
Comment 3: Line 232: The introduction of compatibility evaluation should be placed in the introduction. Here is the introduction of test methods, please adjust it.
Comment 4: Line 189: The mix proportion setting in this paper should describe the reasons for this arrangement and show it in the conclusion analysis, such as why 10 groups of test groups are set and how the components of different materials between each group are set.
Comment 5: Line 590: “It was shown that …and sulphate exposure”, this paper not only studies the influence degree of Geopolymer Materials under sulfuric acid and sulfate conditions, but also some other basic studies, which should be supplemented and expanded appropriately.
Author Response
As attached
